# Influence of Officially Ordered Restrictions during the First Wave of COVID-19 Pandemic on Physical Activity and Quality of Life in Patients after Kidney Transplantation in a Telemedicine Based Aftercare Program—A KTx360° Sub Study

**DOI:** 10.3390/ijerph17239144

**Published:** 2020-12-07

**Authors:** Alexander A. Hanke, Thorben Sundermeier, Hedwig T. Boeck, Elisabeth Schieffer, Johanna Boyen, Ana Céline Braun, Simone Rolff, Lothar Stein, Momme Kück, Mario Schiffer, Lars Pape, Martina de Zwaan, Sven Haufe, Arno Kerling, Uwe Tegtbur, Mariel Nöhre

**Affiliations:** 1Institute of Sports Medicine, Hannover Medical School, 30659 Hannover, Germany; sundermeier.thorben@mh-hannover.de (T.S.); Boeck.Hedwig@mh-hannover.de (H.T.B.); schieffer.elisabeth@mh-hannover.de (E.S.); boyen.johanna@mh-hannover.de (J.B.); braun.celine@mh-hannover.de (A.C.B.); rolff.simone@mh-hannover.de (S.R.); stein.lothar@mh-hannover.de (L.S.); kueck.momme@mh-hannover.de (M.K.); haufe.sven@mh-hannover.de (S.H.); kerling.arno@mh-hannover.de (A.K.); tegtbur.uwe@mh-hannover.de (U.T.); 2Medical Clinic 4: Nephrology and Hypertension, University Hospital Erlangen, 91054 Erlangen, Germany; mario.schiffer@uk-erlangen.de; 3Department of Pediatric Nephrology, University Hospital Essen, 45122 Essen, Germany; lars.pape@uk-essen.de; 4Department of Psychosomatic Medicine and Psychotherapy, Hannover Medical School, 30625 Hannover, Germany; dezwaan.martina@mh-hannover.de (M.d.Z.); noehre.mariel@mh-hannover.de (M.N.); 5Institute of Sports Science, Leibniz University Hannover, 30167 Hannover, Germany

**Keywords:** COVID-19, restrictions, physical activity, quality of life, QoL, kidney transplantation

## Abstract

Guidelines recommend a healthy lifestyle and regularly physical activity (PA) after kidney transplantation (KTx). The KTx360° program is a multicenter, multisectoral, multimodal, telemedicine-based follow-up care program. Effects of the first COVID-19 wave restrictions on health-related quality of life and PA of supervised KTx360° patients were evaluated using an online questionnaire. Six hundred and fifty-two KTx360° patients were contacted via email and were asked to complete the Freiburg questionnaire of physical activity and the Short form 12 Health Survey (SF-12) online. Pre-pandemic and lockdown data were compared in 248 data sets. While sporting activity decreased during the COVID-19 pandemic, basic and leisure activity increased, resulting in increased overall activity. The physical component scale of the SF-12 was in the low normal range before as well as during the pandemic, with a small but significant increase during the pandemic. The mental component scale showed normal values before and during pandemic with a small but statistically significant decrease. Our study supports the effectiveness of a telemedicine based program for KTx patient care in maintaining PA and quality of life during the first peak of the COVID-19 pandemic. However, further research and observation during the ongoing pandemic are required.

## 1. Introduction

Chronic kidney disease is a common disease worldwide with a prevalence rate of 7.2% in adults ≥30 years of age and 23.4–35.8% in adults ≥64 years of age [1]. Kidney transplantation (KTx) is widely considered the treatment of choice for patients with end-stage renal disease and is associated with improved morbidity, mortality and quality of life (QoL) compared to other kidney replacement procedures [2]. 

The guideline for the care of KTx recipients recommends adopting a healthy lifestyle with regular physical activity (PA) [3]. It has been shown that PA post KTx can reduce the risk for developing diabetes or cardiovascular diseases, and even mortality [4]. Effectivity of exercise programs in increasing performance has been shown in a recent review including a total of 24 studies [5]. 

The KTx360° program is a multicenter, multisectoral, multimodal, telemedicine based follow-up care program. Besides nephrological care, close psychosomatic and sports medical support are combined to improve care and reduce healthcare costs after KTx [6]. One major aspect is the implementation of telemedicine supported training based on individual developed training plans. 

In the beginning of 2020, the COVID-19 pandemic forced governments worldwide to establish drastic measures in order to stop the spreading of the virus. The restrictions in Germany included social distancing, namely the avoidance of close contact with other humans whenever possible. In this context, gyms and sport clubs as well as rehabilitation units were temporarily closed. In Germany, people were allowed to leave the house for individual PA.

The aim of our study was to evaluate the effects of COVID-19 pandemic restrictions on health related QoL and PA of telemedicine-based supervised KTx360° patients using an online questionnaire.

## 2. Materials and Methods 

The study was approved by the local ethics committee and was conducted in accordance with the Helsinki Declaration and European Union’s Convention on Human Rights and Biomedicine. All participants gave their written informed consent before participating in the KTx360° program.

### 2.1. Study Population

Participants were recruited within the structured post KTx program KTx360° [6]. To date, 825 patients have been screened for participation within this ongoing interdisciplinary post-transplant program for a structured interventional PA program supervised by the Institute of Sports Medicine of Hannover Medical School. KTx360° participants were contacted via telephone by employees of the nephrological, sportsmedical, and psychosomatic departments. Each patient had at least one study independent remote contact during the pandemic restrictions for regular KTx360° care. Patient´s charts were screened for complete pre-pandemic information on the questionnaires (SF-12 and “Freiburg questionnaire of physical activity” (FrQ)). At the end of the restriction period, 652 participants were contacted via e-mail. The participants were asked to complete an online form which was hosted on a secure server harbored at Hannover Medical School, Germany. A software tool for questionnaires in social sciences, namely the SoSci-Survey tool (SoSci Survey GmbH, Munich, Germany), was used. 

### 2.2. Questionnaire

The online questionnaire included following different sections which are described below in detail:Freiburg questionnaire of physical activity (FrQ)Short form 12 Health Survey (SF-12)

### 2.3. FrQ of Physical Activity

The questionnaire was developed and validated by Frey and coworkers and is described elsewhere in detail [7]. In brief, it is used to assess all kinds of everyday activities (work and leisure) as well as sporting activities of the last week, respectively, the last month. It consists of eight sets of questions, each based on the type, duration, and intensity of activity, and specified the PA as metabolic equivalents of task (MET)-hours per week (MET*h*week^−1^) in three section scores: basic, leisure time, and sporting activity. Basic activities include everyday routes on foot or by bicycle, gardening, and climbing stairs. Leisure activities include walks, bike rides, bicycle ergometer training, dancing, and bowling. Sporting activity includes information on all different sports including swimming. 

Furthermore, a general score of all section scores is calculated. Thus, results can be categorized into three groups based on general scores: “sufficiently active” (≥30 MET*h*week^−1^), “minimum requirement fulfilled” (<30 and ≥14 MET*h*week^−1^) and “far too little active” (<14 MET*h*week^−1^). Furthermore, a sporting activity score greater than 14 MET*h*week^−1^ automatically assorts into “sufficient active”.

### 2.4. SF-12

The SF-12 is an instrument for recording the generic health-related QoL [8]. It has been validated and described in various studies [9,10,11,12,13]. In brief, it consists of 12 questions resulting in two scale scores: the Physical Component Summary (PCS-12) and the Mental Component Summary (MCS-12). Both are standardized combined scores with a mean of 50 and a standard deviation of 10 based on data from the US general population with the higher score indicating better QoL.

### 2.5. Time Points

The time points were defined as followed:T1—Pre COVID-19 pandemic. Data assessed before the lockdown. T1 visits used here were their regular KTx360° visits closest before pandemic lockdown.T2—COVID-19 lockdown. Data were assessed during the lockdown in May 2020.

### 2.6. Statistical Analysis

All data are shown as means (±standard deviation, SD). Normal distribution was verified by a Kolmogorov–Smirnov test. For a comparison between time points (T1 and T2), a two-sided student´s t-test was used with Cohen’s d as the effect size. Thereby, a d between 0.2 and 0.5 stands for a low, a d between 0.5 and 0.8 for a moderate, and a d over 0.8 for a high effect. Furthermore, a chi-square test and a Wilcoxon signed-rank test were performed for comparison of FrQ activity categorization. Categorization changes are displayed in a Sankey Plot. Data were analyzed using SPSS for Windows (SPSS 27, IBM, New York City, NY, USA), and GraphPad Prism (Version 6.01, GraphPad Software Inc., San Diego, CA, USA). We considered *p* < 0.05 as statistically significant.

## 3. Results

### 3.1. Participants

Six hundred and fifty-two KTx360° patients were contacted via email and 248 returned completed data sets. Eighty-nine (35.9%; age 52.3 ± 13.7 years) participants were female while 159 (64.1%; age 56.3 ± 13.0 years) were male. Time since transplantation was 7.6 ± 6.0 years (range 0.5–34.9 years).

### 3.2. FrQ Results

Detailed data on FrQ results are displayed in Table 1. While during lockdown the sporting activity significantly decreased, the basic activity as well as the leisure activity increased, and thus total activity also increased.

Details on PA categorizations are displayed in Table 2. PA categorization was also significantly affected. While the majority (n = 136) had no change in categorization, the categorization improved in 80 patients while there was a decrease in 32 patients, only (*p* < 0.001). Changes in categorization are displayed in Figure 1.

### 3.3. SF-12 Results

Twelve SF-12 datasets had to be sorted out due to missing data at T1. Thus, a total of 236 patients were included into the SF-12 analysis. The SF-12 results are displayed in Table 3. The physical component score (PCS-12) of the SF-12 was lower compared to the mental component score (MCS-12). The PCS-12 increased and the MCS-12 decreased significantly during lockdown. However, both values were well within the standard deviation of 10 and the MCS-12 close to the norm at both time points.

## 4. Discussion

While the second peak wave of the COVID-19 pandemic is still growing at the moment, lessons learned from the first wave have to be considered as essential knowledge. Our study supports the effectiveness of a telemedicine-based program for KTx patient care in maintaining PA and QoL during the first peak COVID-19 pandemic.

As stated previously by others, PA was significantly influenced by COVID-19 pandemic associated lockdown measures [14,15,16,17,18]. The multidisciplinary, multimodal, and telemedicine-based KTx360° program approach might have led to the observed stabilization and even an increase in PA in the KTx360° participants during the lockdown. While, specifically, sports activity decreased, most likely due to closed gyms and rehabilitation units, KTx360° patients were advised and motivated by physicians, mental health professionals, and sports scientists to increase their basic and leisure activities, such as going for a walk or riding a bicycle. Thus, leisure activities increased in the study population. This constitutes a contrast to previously reported PA reduction in chronically ill populations during the COVID-19 pandemic. Namely, it has been described for patients with Parkinson’s [19] and neuromuscular diseases [16], chronic coronary syndrome [20], cystic fibrosis [21], and chronic kidney disease with necessity of hemodialysis [22].

Our data show statistically significant differences over time in Qol as shown by SF-12 data. The PCS-12 was 43.3 (±10.6) pre-pandemic and therefore below the standardized mean score of 50, which can be explained by the underlying kidney disease and comorbidities. The pandemic restrictions did not have further impact on physical aspects of QoL. A slight but statistically significant improvement could be observed, which however does not correspond to an improvement of clinical importance. Regarding mental aspects of QoL, participants showed a slight and statistically significant decline in the MCS-12, while they were still reporting good mental QoL with a mean value of 49.4 (±10.4). In a few publications an effect of the pandemic on mental health has been discussed before [23,24]. In this context especially patients with elevated risks such as comorbidities, immunosuppression, higher age, and others are named to be more sensitive to mental health issues caused by the COVID-19 pandemic [23]. Thus, a more intense psychosocial assessment and monitoring for such patients at risk seems necessary [23]. Thus, we hypothesize that the multimodal approach in the KTx360° program might have been helpful to stabilize the mental status, even in vulnerable patients and at dramatic times.

The American Center for Disease Control and Prevention classified patients after KTx as high-risk group for experiencing a severe course of COVID-19 [25]. When compared to the general population, a significantly increased fatality rate in transplanted patients has been described recently [26,27]. However, in a review study on COVID-19 in KTx recipients, the general risk of COVID-19 infection was not elevated, while a higher risk for a more severe course of COVID-19 can be detected, especially in the context of existing comorbidities [28]. Thus, recommendations for increasing activities while maintaining social distance seem advisable to keep up PA levels, and following the stabilization of QoL.

In a few publications, telehealth aspects during COVID-19 pandemic were addressed before. In order to maintain medical care effectively while establishing social distance, an increase in the use of telemedicine is required [29]. Thus, it has been described that, in different countries and medical settings, telemedicine has already gained importance during the pandemic [30,31,32]. However, telemedicine itself is connected to some major administrative challenges such as training of the doctors for virtual care delivery, integration of telemedicine services into existing medical documentation systems, and finally the willingness and ability of the patients to accept such a proceeding [33,34]. While, in the past, especially the elderly were not capable of using the necessary technologies, recent data show that increasing technology knowledge and handling can be recognized in this population, and thus telemedicine gains broader acceptance and satisfaction for both, physicians and patients [35]. However, the KTx360° project was introduced in 2017 [6] and participating patients as well as the staff were used to being in contact via telephone or online video calls way before the 2020 pandemic. Thus, the acceptance of telemedicine contacts was possibly higher.

In pre-pandemic times, patients in the KTx360° program were, beside telemedicine contacts as described above, also personally supervised during local visits in the hospital. The aim of these visits was the analysis of individual PA status and instruction following for individual exercises and internalization of possibilities to increase everyday activities. However, during lockdown, the personal in-house contacts were completely interrupted for sports medical as well as psychosomatic purposes. In order to give general recommendations for increasing PA with special attention to the pandemic situation, we launched a special Youtube channel with short educative clips transporting instructions for specific home workouts and activity proposals to the patients in an entertaining way. Additionally, we strongly recommended increasing basic and leisure activities whenever possible, especially when sporting activities are reduced due to closed sporting facilities (e.g., “take the bicycle instead of public transportation or a car” or “try to go for a walk everyday”). Furthermore, wearables including a daily step counter seemed to be of great use for the determination of individual activity goals. This is in accordance with a recently published study which stated that the use of smartphone apps can improve PA in individuals during a lockdown [17].

### Limitations

There are some limitations to consider when interpreting the data of our study. First, we were able to include only 38% of the KTx360° patients into the study. However, the beneficial effects of the telemedicine-based program in our high-risk cohort are still consistent, and thus considerable.

Data for T1 were used from regular KTX360° visits closest to the pandemic lockdown. Since T2 data were assessed in May 2020 and pre-pandemic T1 data before March 2020 a seasonal interference with assessed PA might be possible. Furthermore, our results have not been normalized for the degree of the teletraining support to the patients since effectivity of encouragement for increasing PA is difficult to be quantified. Thus, the offered follow up might show differences within the study group with a possible effect on effectivity of the program.

Furthermore, it is important to keep in mind that QoL as well as PA might have been affected by factors not examined in this study. Therefore, it is important to interpret these data with caution.

## 5. Conclusions

Our study demonstrates that we were able maintain PA and QoL in our telemedicine-based program for KTx patient care during the first peak of the COVID-19 pandemic. While sporting activity decreased, the overall activity level increased, mostly caused by an increase in everyday and leisure activity. Furthermore, health-related QoL as assessed by the SF-12 questionnaire was close to population norms with physical QoL being somewhat lower than mental QoL, which can be explained in a group of physically ill patients. However, QoL did not show clinically relevant differences between time points, accounting for a stabilized QoL in KTx patients during lockdown. However, while the COVID-19 pandemic is still ongoing, further observation seems necessary to substantiate the promising results.

## Figures and Tables

**Figure 1 ijerph-17-09144-f001:**
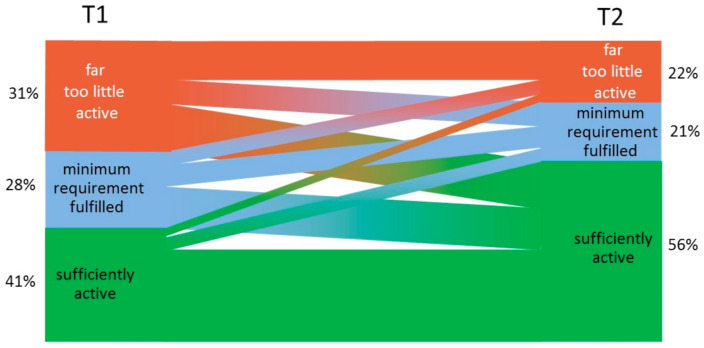
Sankey Plot of PA categorization changes in the Freiburg Questionnaire of Physical Activity. T1 described categorization before COVID-19 pandemic, while T2 describes PA during the first lockdown in May 2020.

**Table 1 ijerph-17-09144-t001:** FrQ results. Results display MET*h*week^−1^ depending on dimensions of FrQ. Time points are T1: before COVID-19 pandemic; T2: during lockdown.

FrQ Dimension (MET*h*week^−1^ (mean ± SD))	T1	T2	t-test	Cohen’s d
Basic score	13.0 ± 15.4	21.7 ± 25.6	t =−5.774, *p* < 0.001	1.92
Leisure score	10.4 ± 12.3	16.2 ± 22.0	t =−4.007, *p* < 0.001	1.40
Sport score	8.3 ± 19.0	5.0 ± 10.2	t = 2.684, *p* = 0.008	0.86
Total score	31.7 ± 31.0	43.0 ± 37.1	t = −4.507, *p* < 0.001	1.54

**Table 2 ijerph-17-09144-t002:** FrQ Categorization of Physical Activity. Results display categorization of PA sufficiency in FrQ. Time points are T1: before COVID-19 pandemic; T2: during lockdown. (Wilcoxon *p*-value: *p* < 0.001).

FrQ Categorization of Physical Activity	T1 [n (%)]	T2 [n (%)]
far too little active	78 (31%)	55 (22%)
minimum requirement fulfilled	69 (28%)	53 (21%)
sufficiently active	101 (41%)	140 (56%)

**Table 3 ijerph-17-09144-t003:** SF-12 results. Results display dimensions of SF-12. PCS-12 dimension represents physical aspects of QoL, while MCS-12 represents mental aspects of QoL.

SF-12 Dimension (mean ± SD)	T1	T2	t-test	Cohen’s d
PCS-12	43.3 ± 10.6	44.5 ± 10.5	t = −0.208, *p* = 0.045	0.37
MCS-12	50.7 ± 10.3	49.2 ± 10.4	t = 2.396, *p* = 0.017	0.47

Time points are T1: before the COVID-19 pandemic; T2: during lockdown.

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
