# Peer review of "Influence of Officially Ordered Restrictions during the First Wave of COVID-19 Pandemic on Physical Activity and Quality of Life in Patients after Kidney Transplantation in a Telemedicine Based Aftercare Program—A KTx360° Sub Study"

_ijerph, 2020, doi:10.3390/ijerph17239144_

Round 1
Reviewer 1 Report
Hanke et al., report results from a prospective cohort and kidney transplant recipients following physical activity prior to and during the COVID-19 pandemic. Interestingly, these patients showed an increase in overall physical activity during the pandemic—but if I’m understanding correctly—it was an increase in leisure time/routine physical activity and a decrease in deliberate exercise. The study is well-written, the results are intriguing, but I have a few questions. Mostly, on if the results are just due to seasonality, the more specifics on what types of activities increased/decreased during the pandemic. I also have some minor suggestions to improve the presentation of the results.
- If I’m understanding correctly, these patients were all previous enrolled in the KTx360 program and were already being followed and given a structured lifestyle intervention via telemedicine? It’s telemedicine the whole time Is there any reason we’re only looking at one single pre-COVID time-point? An interrupted time-series using all assessments would be interesting. I’d be most interested in T1 comparisons in May 2019 to control for seasonality. Again—I’m assuming the T1 visits used here were their follow-up visits closest to the pandemic before lockdowns? If so, I’m imagining physical activity would usually increase from Jan-Mar vs. May (I’ve never been to Hanover—in the Spring or the Winter--but I’m assuming it gets cold and people are less likely to be active?)
- Could you give a bit more detail on the differences between the basis, leisure, and sport scores? Apologies—I couldn’t find an English version of the 1999 paper. I’m curious what kinds of activities people are doing more of. I’m assuming running /jogging/cycling is all under “sport”? If it’s driven by dog walking or gardening—again—I worry about seasonality. Any insights you can give from the item-level responses would be extremely helpful here. Because honestly—if I’m understanding the domains correctly—I might have suspected the opposite. That perhaps people would be more likely to spend more time deliberately exercising (i.e., “sport”) perhaps because they spend less time commuting and less time being physically active as part of their normal day to day routine.
- Could you add percentages to Table 2?
FrQ Categorization of Physical Activity |
T1 N (%) |
T2 N (%) |
far too little active |
78 (31) |
55 (22) |
minimum requirement fulfilled |
69 (28) |
53 (21) |
sufficiently active |
101 (41) |
140 (56) |
- When comparing the continuous scores, you’re using a paired t-test—which is great because you’re comparing how much each individual changed. When comparing the categories in table 2 though—you’re just a chi-square which is not looking at changes within individuals—it’s just looking at the marginal distributions. Suggest using either a marginal homogeneity test or a Wilcoxon signed-rank test instead (though FYI—the Wilcoxon approach will be treating this as an ordinal outcome. I think that’s OK though).
- Honestly—to my point above though—the p-value will still likely be statistically significant. I think it would be more helpful to also show information about the changes. Could you make a Sankey plot? Or just a contingency table of T1 vs. T2 would be nice. Basically, any way of showing how many individuals transitioned from one category to another.
- Related—could you provide the distributions of the subject-specific changes in Tables 1 and Table 3? The p-values in Table 1 are saying the average patient is making a change, but are all making small increases? Some making large increases and others making decreases? Suggest including a plot if possible—but I realize a spaghetti plot with 248 will likely be indecipherable.
- Are there any predictors of if your scores increased or decreased?
- Do you have any other information on these participations beyond age, and sex? Years since transplant would be interesting.
- Minor: line 146: suggest “COVID-19 pandemic” instead of “Corona pandemic”.
- Lines 195-200 at the end of the discussion: were these new parts of the intervention added during the pandemic? I would suggest being extremely clear here and perhaps describing anything like this in the methods that changed in response to the pandemic. One could argue that the YouTube series and wearable devices could have improved physical activity in the covid-free world as well.
- Sorry this is going to come off as rude, but I would suggest toning down the conclusion that it’s really your telemedicine intervention that’s maintaining/increasing physical activity. Again—if you had switched to telemedicine or changed something significant about your program—that might be different—but I think what you have here are results on how the pandemic has impacted individual human behavior (though still interesting!)
Author Response
First of all, we would like to thank the reviewer for spending time in reviewing our manuscript. We found your comments very interesting and think that changes according to your comments improved the manuscript severely.
Here are our answers to your queries:
- If I’m understanding correctly, these patients were all previous enrolled in the KTx360 program and were already being followed and given a structured lifestyle intervention via telemedicine? It’s telemedicine the whole time Is there any reason we’re only looking at one single pre-COVID time-point? An interrupted time-series using all assessments would be interesting. I’d be most interested in T1 comparisons in May 2019 to control for seasonality. Again—I’m assuming the T1 visits used here were their follow-up visits closest to the pandemic before lockdowns? If so, I’m imagining physical activity would usually increase from Jan-Mar vs. May (I’ve never been to Hanover—in the Spring or the Winter--but I’m assuming it gets cold and people are less likely to be active?)
Answer to the Reviewer: That is definitely a good point. Since this is a sub study out of the KTX360°-Project we are – due to legal contracts made before the start of the project - not allowed to publish more data out of the project before the main paper is published. However, since we understand your concerns on seasonal effects we added changes to the material and limitation section. Furthermore, we added the hint to the fact of being a sub study in the title.
Following changes were made:
P1L6: "- a KTX360° sub study"
P4L107: "T1 visits used here were their regular KTX360° visits closest before pandemic lockdown."
P7L211: "Data for T1 were used from regular KTX360° visits closest to the pandemic lockdown. Since T2 data were assessed in May 2020 and prepandemic T1 data before March 2020 a seasonal interference with assessed PA might be possible."
- Could you give a bit more detail on the differences between the basis, leisure, and sport scores? Apologies—I couldn’t find an English version of the 1999 paper. I’m curious what kinds of activities people are doing more of. I’m assuming running /jogging/cycling is all under "sport"? If it’s driven by dog walking or gardening—again—I worry about seasonality. Any insights you can give from the item-level responses would be extremely helpful here. Because honestly—if I’m understanding the domains correctly—I might have suspected the opposite. That perhaps people would be more likely to spend more time deliberately exercising (i.e., "sport") perhaps because they spend less time commuting and less time being physically active as part of their normal day to day routine.
Answer to the Reviewer: We added to the methods section more detailed information on the Freiburg Questionnaire: "In brief, it is used to assess all kind of everyday activities (work and leisure) as well as sporting activities of the last week, respectively the last month. It consists of eight sets of questions, each based on the type, duration and intensity of activity, and specified the PA as metabolic equivalents of task (MET)-hours per week (MET*h*week-1) in three section scores: basic, leisure time, and sporting activity. Basic activities include everyday routes on foot or by bicycle, gardening and climbing stairs. Leisure activities include walks, bike rides, bicycle ergometer training, dancing, and bowling. Sporting activity includes information on all different sports including swimming."
Furthermore, we added to the discussion section a more specific description, how the basis and leisure activity increase was forced: "Additionally, we strongly recommended increasing basis and leisure activities whenever possible – especially when sporting activities were reduced due to closed sporting facilities (e.g. "take the bicycle instead of public transportation or a car" or "try to go for a walk everyday")." P7L209ff
- Could you add percentages to Table 2?
FrQ Categorization of Physical Activity |
T1 N (%) |
T2 N (%) |
far too little active |
78 (31) |
55 (22) |
minimum requirement fulfilled |
69 (28) |
53 (21) |
sufficiently active |
101 (41) |
140 (56)
|
Answer to the Reviewer: As recommended we added the percentages to the table. See P5L136…
- When comparing the continuous scores, you’re using a paired t-test—which is great because you’re comparing how much each individual changed. When comparing the categories in table 2 though—you’re just a chi-square which is not looking at changes within individuals—it’s just looking at the marginal distributions. Suggest using either a marginal homogeneity test or a Wilcoxon signed-rank test instead (though FYI—the Wilcoxon approach will be treating this as an ordinal outcome. I think that’s OK though).
Answer to the Reviewer: We calculated as desired a Wilcoxon signed-rank test and found p<0.001. We added this to the manuscript. (see P4L118 & P5L150)
- Honestly—to my point above though—the p-value will still likely be statistically significant. I think it would be more helpful to also show information about the changes. Could you make a Sankey plot? Or just a contingency table of T1 vs. T2 would be nice. Basically, any way of showing how many individuals transitioned from one category to another.
Answer to the Reviewer: We found the idea of adding a Sankey plot wonderful and inserted it into the manuscript (see Fig. 1)
- Related—could you provide the distributions of the subject-specific changes in Tables 1 and Table 3? The p-values in Table 1 are saying the average patient is making a change, but are all making small increases? Some making large increases and others making decreases? Suggest including a plot if possible—but I realize a spaghetti plot with 248 will likely be indecipherable.
Answer to the Reviewer: see above
- Are there any predictors of if your scores increased or decreased?
Answer to the Reviewer: Since – as stated before – this is a substudy out of the KTx360° project and we are forced not to be too deep with the description of the primary population and its outcome we did not calculate predictors. Such data will be published in the overall KTx360° publication. We are sorry to do so, but cannot change it at the moment.
- Do you have any other information on these participations beyond age, and sex? Years since transplant would be interesting.
Answer to the Reviewer: As desired we added: "Time since transplantation was 7.6±6.0 years (range 0.5-34.9 years)." See P6L150…P4L121
- Minor: line 146: suggest "COVID-19 pandemic" instead of "Corona pandemic".
Answer to the Reviewer: A true mistake … sorry for that. It has been replaced. See P6L150…
Lines 195-200 at the end of the discussion: were these new parts of the intervention added during the pandemic? I would suggest being extremely clear here and perhaps describing anything like this in the methods that changed in response to the pandemic. One could argue that the YouTube series and wearable devices could have improved physical activity in the covid-free world as well.
Answer to the Reviewer: Thank you for this specific comment on out changed intervention. We completely agree that it has to be described more clearly. Thus, we added the following to the discussion section:
P7L179ff: However, the KTx360° project was introduced in 2017 [6] and participating patients and as well as the staff were used to be in contact via telephone or online video calls way before the 2020 pandemic. Thus, the acceptance of telemedicine contacts was possibly higher.
In prepandemic times patients in the KTx360° program were – beside telemedicine contacts as described above – also supervised personally during local visits in the hospital. Aim of these visits was analysis of individual PA status and following instruction for individual exercises and internalization of possibilities to increase everyday activities. However, during lockdown the personal in house contacts were completely interrupted for sportsmedical as well as psychosomatic purposes. In order to give general recommendations for increasing PA with special attention to the pandemic situation we launched a special Youtube channel with short educative clips transporting instructions for specific home workouts and activity proposals to the patients in an entertaining way. Additionally, we strongly recommended increasing basis and leisure activities whenever possible – especially when sporting activities were reduced due to closed sporting facilities (e.g. "take the bicycle instead of public transportation or a car" or "try to go for a walk everyday").
- Sorry this is going to come off as rude, but I would suggest toning down the conclusion that it’s really your telemedicine intervention that’s maintaining/increasing physical activity. Again—if you had switched to telemedicine or changed something significant about your program—that might be different—but I think what you have here are results on how the pandemic has impacted individual human behavior (though still interesting!)
Answer to the Reviewer: First of all, it doesn´t sound rude at all. Science is always a matter of discussion and keeps the distance to the conspiracy theorists that are stating strange opinions in our times … We changed the sentence in the conclusions to (P7L222ff): "Our study demonstrates that we were able maintain PA and QoL in our telemedicine-based program for KTx patient care during the first peak of the COVID-19 pandemic." We hope that this toning finds your acceptance.
Thank you again and a happy holiday season!
Reviewer 2 Report
Congratulations for a well organized study that you have completed and you are presenting with your submitted manuscript.
However I have some concerns reviewing your work which are the following:
I have not any information regarding the calculation of the power of the study in order your statistical evaluation to be objectively confirmed with an accepted sample size.
I would also like to know if your results have been normalized for the degree of the teletraining support to the patients while in lines 74-75 you are station that "each patient had at least one study independent remote contact...". Thus a multifactorial analysis would be then more relevant because either physical activity at home or the benefits of home exercising could be different and analogous to the offered follow up. In any other case the limitations of the study should be then rephrased including that information.
I would also like to know how far from the lockdown period the survey performed. There is always the case the patient's responses might not be equal if some people answered close and some months later from the time full of restrictions. If does exist such a case then it would also has to be mentioned in the study limitations
Author Response
First of all, we would like to thank the reviewer for spending time in reviewing our manuscript. We found your comments very interesting and think that changes according to your comments improved the manuscript severely.
Here are our answers to your queries:
Congratulations for a well organized study that you have completed and you are presenting with your submitted manuscript.
Answer to the Reviewer: Thank you very much!
However I have some concerns reviewing your work which are the following:
I have not any information regarding the calculation of the power of the study in order your statistical evaluation to be objectively confirmed with an accepted sample size.
Answer to the Reviewer: Since this is a sub study out of a continuing project (KTx360°) we did not calculate a power analysis a priori. However, to meet your point we added Cohen’s d as effect size to estimate the strength of the calculated data. We hope to find your understanding and acceptance.
I would also like to know if your results have been normalized for the degree of the teletraining support to the patients while in lines 74-75 you are station that "each patient had at least one study independent remote contact...". Thus a multifactorial analysis would be then more relevant because either physical activity at home or the benefits of home exercising could be different and analogous to the offered follow up. In any other case the limitations of the study should be then rephrased including that information.
Answer to the Reviewer: This is definitely a good point. Unfortunately we were not able to quantify the effectivity of the remote contacts. To meet your comment we added to the limitations section: "Furthermore, our results have not been normalized for the degree of the teletraining support to the patients since effectivity of encouragement for increasing PA is difficult to be quantified. Thus, the offered follow up might show differences within the study group with a possible effect on effectivity of the program. " (P7L222ff)
I would also like to know how far from the lockdown period the survey performed. There is always the case the patient's responses might not be equal if some people answered close and some months later from the time full of restrictions. If does exist such a case then it would also has to be mentioned in the study limitations
Answer to the Reviewer: As we described in the methods section, the survey was performed in May 2020 during the ending lockdown period. (see P4L109). Thus, the information given in all questionnaires were fresh.
Thank you again for reviewing and a happy holiday season!!!